Active ingredients and molecular targets of Taraxacum mongolicum against hepatocellular carcinoma: network pharmacology, molecular docking, and molecular dynamics simulation analysis

Zheng Yanfeng
Ji Shaoxiu
Li Xia
Feng Quansheng fengqscdutcm@126.com
Basic Medical College, Chengdu University of Traditional Chinese Medicine , Chengdu , Sichuan , China
Adnan Mohd
Electronic publication date: 2022 Jul 18
Publication date: 2022
Volume: 10
Electronic Location ID: e13737
Received 2022 Mar 25; Accepted 2022 Jun 25
Copyright: ©2022 Zheng et al.
Copyright year: 2022
Copyright holder: Zheng et al.
License: This is an open access article distributed under the terms of the Creative Commons Attribution License, which permits unrestricted use, distribution, reproduction and adaptation in any medium and for any purpose provided that it is properly attributed. For attribution, the original author(s), title, publication source (PeerJ) and either DOI or URL of the article must be cited.
License URL: https://creativecommons.org/licenses/by/4.0/

Keywords: Taraxacum Mongolicum, Hepatocellular carcinoma, Network pharmacology, Molecular docking, Molecular dynamics simulation

Funding: The Sichuan Province Key R&D Plan Project 2020YFS0301 The Sichuan Province Applied Basic Research Project 2021YJ0253 This work was supported by the Sichuan Province Key R&D Plan Project (No. 2020YFS0301) and the Sichuan Province Applied Basic Research Project (2021YJ0253). The funders had no role in study design, data collection and analysis, decision to publish, or preparation of the manuscript.

==============================
Background

Taraxacum mongolicum (TM) is a widely used herb. Studies have reported that TM exhibits growth-inhibitory and apoptosis-inducing on multiple tumors, including hepatocellular carcinoma (HCC). The active ingredients, targets, and molecular mechanisms of TM against HCC need to be further elucidated.

Methods

We identified the active ingredients and targets of TM via HERB, PubChem, SwissADME, SwissTargetPrediction, and PharmMapper. We searched HCC targets from GeneCards, Comparative Toxicogenomics Database (CTD), and DisGeNET. Then, the intersection of drug targets and disease targets was uploaded to the STRING database to construct protein-protein interactions (PPI) networking whose topology parameters were analyzed in Cytoscape software to screen hub targets. Next, we used Metascape for Gene Ontology (GO) and Kyoto Encyclopedia of Genes and Genomes (KEGG) enrichment analysis, and we employed AutoDock vina, AMBER18 and PyMOL software along with several auxiliary tools for molecular docking and molecular dynamics (MD) simulation. Finally, based on the in silico findings, cellular experiments were conducted to investigate the effect of TM on HSP90AA1 gene expression.

Results

A total of 228 targets and 35 active ingredients were identified. Twenty two hub targets were selected through PPI networking construction for further investigation. The enrichment analysis showed that protein kinase binding, mitogenactivated protein kinase (MAPK) and phosphatidylinositol 3-kinase (PI3K)/Akt signaling pathways were mainly involved. Molecular docking and MD simulation results supported good interaction between HSP90 protein and Austricin/Quercetin. The in vitro assay showed that TM inhibited the proliferation of HepG2 cells and the expression of HSP90AA1 gene.

Conclusions

This study is the first to use network pharmacology, molecular docking, MD simulation and cellular experiments to elucidate the active ingredients, molecular targets, and key biological pathways responsible for TM anti-HCC, providing a theoretical basis for further research.

Introduction

Liver cancer is the second leading cause of cancer-related deaths globally, remains a global health challenge (Sung et al., 2021). The epidemiological situation of liver cancer in China is worse than in other countries worldwide. More than half of all deaths and newly diagnosed cases are in China (Sung et al., 2021). Surveillance Epidemiology End Results has reported that liver cancer is the fastest increasing cause of cancer-related deaths in the USA since the early 2000s, and it is estimated that, by 2030, liver cancer is projected to become the third leading cause of cancer-related deaths if these trends continue (Rahib et al., 2014). Hepatocellular carcinoma (HCC), the major pathological type of primary liver cancer, accounts for about 90% of all cases, and the morbidity is on the rise (Liver, 2018). Despite significant improvements in the management of HCC, the prognosis remains bleak. None of the current treatments will be sufficient to considerably decrease the number of deaths due to HCC in the coming decades (Yang et al., 2019). Hepatectomy and liver transplantation are considered potentially curative approaches for HCC. However, only 10% of patients are eligible for resection since most of the patients will present with advanced disease, and the post-resection 5-year recurrence rate of HCC is still statistically as high as 70% (Kulik & El-Serag, 2019). Donor livers for transplantation are scarce, and the problem of post-transplant relapse still exists. Sorafenib, the only currently available standard of systemic therapy for HCC, is limited by its high toxicity and low response rate (Kudo, 2018). Low long-term survival and high recurrence rates in HCC patients continue to be obstacles to surmount. Adjuvant therapies to preclude relapse are an unmet medical need for HCC (Llovet et al., 2021). Developing effective, low-toxicity complementary and alternative drugs for HCC are essential since current treatments fail to achieve satisfactory clinical outcomes.

TM is a widely used herb with the characteristics of extensive pharmacological action, good security, high nutritional and medical value, which is worthy of in-depth study and development application. The theory of traditional Chinese medicine (TCM) holds that TM is bitter and sweet in taste, neutral in property, and effective in clearing liver fire, which is specifically used to treat liver meridian and the corresponding organs. The clinical usage of TM in liver diseases is instinctive and comes from popular knowledge and experience. The first evidence of its therapeutic use is in the Arabian medicine in the X and XI centuries to treat diseases of the liver and spleen (Martinez et al., 2015). The hepatoprotective effects of TM have been identified in liver diseases such as hepatocellular carcinoma (Rehman et al., 2017b), nonalcoholic fatty liver (Davaatseren et al., 2013), and liver failure (Pfingstgraf et al., 2021). In recent years, extensive studies have reported the growth-inhibitory and apoptosis-inducing of TM on multiple tumor cells, including HCC, gastric cancer, lung cancer, and breast cancer (Chien et al., 2018; Deng et al., 2021; Park, Cho & Song, 2014; Rehman et al., 2017a; Ren et al., 2019; Ren et al., 2020; Zhu et al., 2017), but the mechanisms need to be further elucidated.

Based on the theory of system biology, network pharmacology covers high throughput omics data analysis, computer virtual computing, and network database retrieval. It is commonly used in biological systematic network analysis and is a new discipline capable of predicting drug targets from a holistic perspective and increasing the efficiency of drug discovery (Boezio et al., 2017). Network pharmacology breaks the old limitation of ‘one drug, one biological target’ research and is applied widely in active ingredient screening, explanation of effective mechanism, and pathogenesis research (Hopkins, 2008). The systematicness and integrity of network pharmacology coincide with the wholism of TCM, which contribute to understanding the mechanism of multi-ingredient, multi-target, and multi-pathway of herbs (Lai et al., 2020; Zhang et al., 2019). Molecular docking, based on the principle of ligand–receptor interactions, serves as a versatile tool to help understanding how chemical compounds interact with their molecular targets and finds wide application in drug discovery (Ma, Chan & Leung, 2011; Pinzi & Rastelli, 2019). Molecular dynamics (MD) simulation illuminates the dynamic behavior of biomolecules at atomic level with fine quality representation of biomolecules (Vidal-Limon, Aguilar-Toala & Liceaga, 2022).

We applied network pharmacology, molecular docking, MD simulation and cellular experiments to identify active ingredients, molecular targets, and key biological pathways responsible for TM anti-HCC, providing a reference for the follow-up basis research. The detailed workflow is shown in Fig. 1.

Materials & Methods

Active ingredients and predicted targets identification

The chemical components of TM were retrieved from the HERB (http://herb.ac.cn/) database, a high-throughput experiment- and reference-guided database of TCM. HERB not only contains data from SymMap, the Traditional Chinese Medicine Integrated Database (TCMID), the Traditional Chinese Medicine Systems Pharmacology Database and Analysis Platform (TCMSP), and TCM-ID databases, but also inherently adds herbal species and corresponding targets, making it the most comprehensive herbal bioinformatics database now (Fang et al., 2021). The 3D structures of the chemical components downloaded from PubChem (https://pubchem.ncbi.nlm.nih.gov/) (Wang et al., 2017b) were uploaded to SwissADME (http://www.swissadme.ch/) platform for pharmacokinetics and drug-likeness assessment (Daina, Michielin & Zoete, 2017). Among the chemical components with high gastrointestinal absorption, meeting both Lipinski’s rule, and any two of Ghose, Veber, Egan and Muegge filters, were considered as the active ingredients of TM. The predicted targets of the active ingredients were obtained from SwissTargetPredictive (http://www.swisstargetprediction.ch/) platform, PharmMapper Server (http://www.lilab-ecust.cn/pharmmapper/) and HERB database. High levels of the predictive performance of SwissTargetPrediction are based on the similarity principle, through reverse screening, within a sizeable collection of 376342 compounds (Daina, Michielin & Zoete, 2019). PharmMapper Server, an updated integrated pharmacophore matching platform, uses statistical methods for potential target identification, with a database extracted from all the targets in TargetBank, DrugBank, Binding Database, and Potential Drug Target Database (Wang et al., 2017a). All obtained targets were preserved as gene symbols.

Figure 1 Workflow of this study.

Disease targets and intersection targets collection

HCC disease targets were collected from GeneCards (Rappaport et al., 2017) (https://www.genecards.org/), Comparative Toxicogenomics Database (CTD) (Davis et al., 2021) (http://ctdbase.org/), and DisGeNET (Pinero et al., 2020) (https://www.disgenet.org/home/) databases, with the search term “Hepatocellular Carcinoma” “Adult Hepatocellular Carcinoma” and species selected as humans. Only targets marked with “M” or “T” symbols were accepted in the retrieval results of CTD. “M” indicates the target may be a biomarker or play a role in the etiology of HCC, and “T” indicates the target may be a therapeutic target of HCC. The intersection of HCC disease targets and the predicted targets of active ingredients was selected in Venny 2.1.0 (https://bioinfogp.cnb.csic.es/tools/venny/index.html). The intersection targets obtained will be compared with differentially expressed genes in TCGA Liver Cancer Dataset (n = 423, https://tcga.xenahubs.net).

Network construction and analysis

STRING covering 24+ million proteins from 5090 organisms is a database for constructing known and predicted protein-protein interactions (PPI) networks (Szklarczyk et al., 2019). Cytoscape is an authoritative and reliable tool for automated biological analysis and visualization. It can integrate biomolecular interaction networks with high-throughput gene expression data and other molecular state information and is often used to analyze large-scale protein-protein interaction, protein-DNA interaction, and gene interaction (Otasek et al., 2019). The intersection targets were uploaded to STRING11.0 (https://www.string-db.org/) to build a PPI network, with the species limited to “Homo sapiens” and a confidence score ≤ 0.9. Topological parameters of the PPI network were analyzed by the Network Analyzer tool in Cytoscape 3.8.2 software. The higher quantitative values, especially the degree and betweenness centrality in topological parameters, indicate the greater importance of nodes. Targets with the degree and betweenness centrality values above twice the mean were classified as hub targets. Similarly, the interworking network of hub targets and active ingredients was constructed and analyzed by Cytoscape. Active ingredients with the degree greater than the mean were classified as key ingredients.

GO and KEGG enrichment analysis and herb-ingredient-target -pathway network construction

Metascape (https://metascape.org/gp/index.html#/main/step1), one of the effective and efficient tools for functional enrichment analysis, evades confounding data interpretation issues by absorbing most redundancies into representative clusters better than other ways (Zhou et al., 2019). The hub targets were uploaded to Metascape for Gene Ontology (GO) and Kyoto Encyclopedia of Genes and Genomes (KEGG) enrichment analysis to clarify their regulatory pathways and biological functions, with “Homo sapiens” as the analysis species. Terms with a P-value < 0.01, a minimum count over three, and an enrichment factor > 1.5 were considered significant enrichment. The Cytoscape software was used to construct a herb-ingredient-target-pathway network based on KEGG results, and its topological parameters were analyzed with the network analyzer tool therein.

Molecular docking

AutoDock Vina, a new generation of docking software from the Molecular Graphics Lab, generally improves the average accuracy of the binding mode predictions and speeds up operations using a terse scoring function compared to widely-used and most cited AutoDock 4 (Nguyen et al., 2020). AutoDock Vina is a leader in the field of Molecular Docking currently, and its results tend to be better than those obtained from more costly high-throughput experimental screens and are well reproducible (Muegge & Mukherjee, 2016). The semi-flexible molecular docking calculation function of AutoDock Vina software was used to predict the binding affinities between active ingredients and hub targets. Docking results are output in binding free energy. Protein-ligand binding can occur spontaneously only when the system free energy is negative. The lower the energy, the higher the affinity, and the stronger the binding force is between the ingredient and the protein. It is generally accepted that the binding is strong when the binding free energy is less than −5.0 kcal/mol (Hamza et al., 2021). The 3D structures of targets were obtained from the PDB (https://www.rcsb.org/) database before docking. Firstly, ligands, non-protein molecules, and water molecules were removed from the 3D structures by PYMOL2.5.0 software, then hydrogenation, charge addition, and residues repair were conducted in AutoDock Vina, and finally, they were saved in PDBQT format for use as receptors for docking. Likewise, the 3D structures of the active ingredients were converted to PDB format by OpenBabel3.1.1 software, then hydrogenated and energy minimized in AutoDock Vina, and finally, they were transformed into PDBQT format for use as ligands for docking. Molecular docking was carried out at Deepsite (https://www.playmolecule.com/deepsite/) predicted binding sites by AutoDock Vina, and the docking results were visualized by PyMOL.

Molecular dynamics (MD) simulation

The AMBER18 software package was used for MD simulations, and ingredients were parameterized with ANTECHAMBER module and GAUSSIAN09 software (Salomon-Ferrer, Case & Walker, 2013). FF14SB force field was used to process the proteins, and GAFF2 force field was used to process the ingredients. The addition of hydrogen atoms and sodium ions to protein-ingredient complexes was accomplished in LEaP module to ensure overall charge neutrality. Solvation of each complex was performed with the TIP3P water model. The MD simulation was realized with three steps: energy optimization, system equilibrium and dynamics simulation production. Energy optimization was performed using the steepest descent method of 2,500 steps, followed by conjugate gradient method, for 2,500 steps. Position-restrained dynamics simulations (NVT and NPT) were performed at 300 K for 500 ps to achieve system equilibrium. Finally, the system was subjected to a molecular dynamics simulation (NPT) for 100 ns under periodic boundary conditions (PBC), with collision frequency 2 ps−1, system pressure 1 atm, integration step 2 fs, and the trajectory data was saved every 10 ps. Furthermore, the binding free energy of proteins and ingredients was calculated using MM/GBSA.

Cell lines and cell culture

The human HCC cell lines HepG2 were purchased from the Chinese Academy of Sciences. HepG2 cells were cultured in DMEM (Gibco, China) containing 10% FBS (Gibco, Uruguay) and 1% penicillin/streptomycin at 37 °C in the cell incubator with a humid atmosphere containing 5% CO2. Cell operations were performed on an ultra-clean workbench at all times.

Cell counting kit-8 (CCK8) assay

The HepG2 cells were seeded in a 96-well plate at a density of 5 × 103 cells/well after 80% confluence of the recovered cells. Treatment with various concentrations of TM extract (0, 5, 10, 15, and 20 mg/ml) for 24 h in a 5% CO2 incubator at 37 °C after the cells attached to the wall completely. Then 10ul CCK-8 buffer (Dojindo, Japan) was added to each well and incubated for 1 h. A negative control well was set up without seeding cells. Each group was equipped with six auxiliary holes. TM extract (20:1) was purchased from Shaanxi Ivy Bioengineering Co. The absorbance value of each well was measured using a continuous spectrum scanning microplate reader (Molecular Devices, USA) at a wavelength of 450 nm.

Reverse transcription quantitative polymerase chain reaction (RT-qPCR) analysis

FastPure Cell/Tissue Total RNA Isolation Kit V2 (Vazyme, China) was applied for total RNA isolation according to the product instruction. The quality of total RNA was accredited by ScanDrop2 (Analytik, Jena, Germany). One microgram of the total RNA was used for cDNA synthesis following the HiScript III RT SuperMix for qPCR (Vazyme, China) instruction. Measurement of the relative expression of mRNA by RT-qPCR using TB Green® Premix Ex Taq™ II (TAKARA, Japan). β-Actin gene was served as the internal reference and the 2−ΔΔCt method was adopted for the data analysis. The primer sequences of β-Actin and HSP90AA1 were synthesized by Tsingke Biotechnology Co., Ltd (Beijing, China, Table 1).

Table 1 RT-qPCR primer sequences.

Gene		5′to 3′	
HSP90AA1	Forward	GGTCCTGTGCGGTCACTTA	
	Reverse	TTCTGCCTGAAAGGCGAACG	
β-Actin	Forward	CCTTCCTGGGCATGGAGTC	
	Reverse	TGATCTTCATTGTGCTGGGTG	

Statistical analysis

Data were presented as mean ± standard deviation (SD). Statistical analysis was performed using GraphPad Prism 9.0 software. The two-tailed unpaired Student’s t-test was used for comparisons between two groups, the one-way ANOVA was used for comparisons between multiple groups, and the difference was statistically significant at p < 0.05.

Results

Active ingredients and predicted targets of TM

A total of 72 components of TM were searched from HERB database, of which 35 active ingredients were identified by SwissADME (Table 2). The number of targets predicted from SwissTargetPrediction, PharmMapper Server, and HERB for active ingredients was 578, 423, and 417, respectively. A total of 1,126 predicted targets were retained after the duplicates were eliminated.

Table 2 Thirty-five active ingredients of TM.

Herb id	PubChem CID	Active ingredients	SwissADME	
			GI	Lipinski	Ghose	Veber	Egan	Muegge	
HBIN001773	5371272	1,6,6-Trimethyl-7-[(Z)-3-oxobut-1-enyl]-3,8-dioxatricyclo[5.1.0.02,4]octan-5-one	High	Yes	Yes	Yes	Yes	Yes	
HBIN005643	7362	Furfural	High	Yes		Yes	Yes		
HBIN006828	582907	2-t-Butyl-5-methyl-[1,3]dioxolane- 4-carboxylic acid	High	Yes	Yes	Yes	Yes		
HBIN008384	536539	1-(3,7,7-Trimethyl-4-bicyclo [4.1.0] hept-3-enyl)ethanone	High	Yes	Yes	Yes	Yes		
HBIN008798	1183	Vanillin	High	Yes		Yes	Yes		
HBIN010490	19376246	Diacetone alcohol	High	Yes		Yes	Yes		
HBIN010946	551125	5,10-Diethoxy-2,3,7,8-tetrahydro-1H, 6H-dipyrrolo[1,2-a:1′,2′-d]pyrazine	High	Yes	Yes	Yes	Yes	Yes	
HBIN012161	538983	6-Acetyl-4,4,7-trimethylbicyclo[4.1.0] heptan-2-one	High	Yes	Yes	Yes	Yes		
HBIN016904	13864723	Arsanin	High	Yes	Yes	Yes	Yes	Yes	
HBIN016938	101245659	Artecalin	High	Yes	Yes	Yes	Yes	Yes	
HBIN017758	243	Benzoic acid	High	Yes		Yes	Yes		
HBIN019298	689043	Caffeic acid	High	Yes	Yes	Yes	Yes		
HBIN023435	6713966	Austricin	High	Yes	Yes	Yes	Yes	Yes	
HBIN025796	5281416	Esculetin	High	Yes		Yes	Yes		
HBIN025897	5317238	Ethyl caffeate	High	Yes	Yes	Yes	Yes	Yes	
HBIN025971	28310	Ethyl 4-hydroxyphenylacetate	High	Yes	Yes	Yes	Yes		
HBIN033339	5280934	Linolenic acid	High	Yes			Yes		
HBIN035129	689075	Methyl caffeate	High	Yes	Yes	Yes	Yes		
HBIN035816	518900	Methyl 4-hydroxyphenylacetate	High	Yes	Yes	Yes	Yes		
HBIN036159	11005	Myristic acid	High	Yes	Yes		Yes		
HBIN038680	985	Palmitic acid	High	Yes	Yes		Yes		
HBIN039500	999	Phenylacetic acid	High	Yes		Yes	Yes		
HBIN039666	127	4-Hydroxyphenylacetic acid	High	Yes		Yes	Yes		
HBIN039673	54675830	4-Hydroxybenzoate	High	Yes		Yes	Yes		
HBIN039707	10394	3-(4-Hydroxyphenyl)propionic acid	High	Yes	Yes	Yes	Yes		
HBIN040911	8768	3,4-Dihydroxybenzaldehyde	High	Yes		Yes	Yes		
HBIN041495	5280343	Quercetin	High	Yes	Yes	Yes	Yes	Yes	
HBIN043442	5280460	Scopoletin	High	Yes	Yes	Yes	Yes		
HBIN044329	101702520	Sonchuside A	High	Yes	Yes	Yes	Yes	Yes	
HBIN045261	443023	(+)-Syringaresinol	High	Yes	Yes	Yes	Yes	Yes	
HBIN045547	5241825	Taraxacin	High	Yes	Yes	Yes	Yes	Yes	
HBIN045548	9921439	Taraxinic acid	High	Yes	Yes	Yes	Yes	Yes	
HBIN046806	5280535	p-Coumaryl alcohol	High	Yes		Yes	Yes		
HBIN046808	641301	p-Coumaraldehyde	High	Yes		Yes	Yes		
HBIN048047	54690394	D-Ascorbic acid	High	Yes		Yes	Yes		

Disease targets and intersection targets

A total of 1437 HCC disease targets were collected, of which 778 from GeneCards, 509 from CTD, and 150 from DisGENET. After the duplicates were removed, 1,223 targets were obtained. A total of 228 intersection targets were identified by VENNY2.1.0, as shown in Fig. 2. All the intersection targets were among the differentially expressed genes in TCGA Liver Cancer Dataset, see supplementary file.

Figure 2 Intersection targets.

A total of 228 intersection targets were identified by VENNY2.1.0.

PPI network and hub targets

The PPI network was co-constructed by STRING and Cytoscape (Fig. 3), which contains 227 nodes and 5,867 edges, with one dissociative target removed. The average of degree, betweenness centrality, and closeness centrality are 51.6916, 0.0038, and 0.5459, respectively. The node size is proportional to the degree value. Twenty-two targets with degree and betweenness centrality values above twice the mean were categorized into hub targets (Table 3). The color of the hub targets is deepened in the figure.

Figure 3 The PPI network.

The node size is proportional to the degree value. The color of the hub targets is deepened in the figure.

Table 3 Twenty-two hub targets.

NO.	Hub target	Betweenness centrality	Closeness centrality	Degree	
1	GAPDH	0.069492108	0.795774648	168	
2	TP53	0.040729912	0.784722222	165	
3	AKT1	0.039453643	0.771331058	160	
4	MYC	0.02413519	0.736156352	146	
5	ALB	0.044293226	0.736156352	145	
6	EGFR	0.015531628	0.70846395	134	
7	MAPK3	0.01953981	0.70846395	134	
8	VEGFA	0.017537919	0.704049844	133	
9	CASP3	0.011495283	0.704049844	132	
10	STAT3	0.011020537	0.701863354	131	
11	CCND1	0.013690985	0.691131498	128	
12	JUN	0.011489939	0.695384615	128	
13	PTEN	0.013589929	0.68902439	126	
14	EGF	0.014214947	0.691131498	126	
15	IL6	0.01571605	0.691131498	126	
16	MAPK1	0.014283335	0.68902439	125	
17	MAPK8	0.014546931	0.684848485	124	
18	HRAS	0.009985322	0.682779456	122	
19	SRC	0.011244283	0.678678679	121	
20	TNF	0.007928097	0.672619048	117	
21	ESR1	0.012107136	0.664705882	114	
22	HSP90AA1	0.011944643	0.660818713	113	

Ingredient-hub target network and key ingredients

The ingredient–hub target network comprising 58 nodes and 210 edges (Fig. 4) was constructed. The active ingredient nodes are in a circular layout and the hub targets are in a grid layout. The network analysis showed that the average degree value of the ingredients is six. Fourteen active ingredients with the degree values higher than six were classified as key ingredients (Table 4). The blue V-shaped nodes in the figure represent the hub targets.

Figure 4 The ingredient-hub target network.

The active ingredient nodes are in a circular layout and the hub targets are in a grid layout. The blue V-shaped nodes in the figure represent the hub targets.

Table 4 Fourteen key ingredients of TM.

Herb ID	PubChem CID	Key ingredients	Degree	3D structure	
HBIN041495	5280343	Quercetin	18		
HBIN038680	985	Palmitic acid	12		
HBIN033339	5280934	Linolenic acid	12		
HBIN025796	5281416	Esculetin	10		
HBIN044329	101702520	Sonchuside A	9		
HBIN025971	28310	Ethyl 4-hydroxyphenylacetate	9		
HBIN008384	536539	1-(3,7,7-Trimethyl-4-bicyclo [4.1.0] hept-3-enyl)ethanone	8		
HBIN048047	54690394	D-Ascorbic acid	8		
HBIN025897	5317238	Ethyl caffeate	8		
HBIN019298	689043	Caffeic acid	8		
HBIN045548	9921439	Taraxinic acid	7		
HBIN035129	689075	Methyl caffeate	7		
HBIN023435	6713966	Austricin	7		
HBIN008798	1183	Vanillin	7		

GO and KEGG enrichment analysis

Metascape was used to elucidate the biological process (BP), cell composition (CC), and molecular function (MF) annotation of the 22 hub targets. A total of 883 GO terms were obtained, including 807 of BP, 36 of CC, and 40 of MF. The top 20 terms ranked by log (q-value) are shown in Fig. 5. The larger the dot, the more targets are involved. The main biological processes involved in the hub targets are gland development, epithelial cell proliferation, positive regulation of transferase activity, positive regulation of protein phosphorylation, and MAPK cascade. The hub targets widely consist in intracellular and extracellular structures, such as vesicle lumen, membrane raft, membrane microdomain, secretory granule lumen, and cytoplasmic vesicle lumen. Protein kinase binding is an important molecular function of the hub targets since the largest number of targets are involved, and six of the top 20 terms are related to protein kinases. KEGG pathway annotation of the hub targets yielded 217 terms. The top 20 terms ranked by gene ratio are shown in Fig. 5, eight of which pertain to HCC, including pathways in cancer, proteoglycans in cancer, mitogen-activated protein kinase (MAPK), phosphatidylinositol 3-kinase (PI3K)/Akt signaling pathway, epidermal growth factor receptor (EGFR) tyrosine kinase inhibitor resistance, foxo signaling pathway, ErbB signaling pathway, and MicroRNAs in cancer. The pathways in cancer and proteoglycans in cancer are the terms that involve the highest number of targets. Inspection of the detailed pathway information revealed that these two terms contain multiple pathways, such as the MAPK signaling pathway, the PI3K/AKT signaling pathway, the janus kinase/signal transducer and activator of transcription (JAK/STAT) signaling pathway, and hypoxia-inducible factor-1 (HIF-1) signaling pathway, where most of the targets are located in the MAPK signaling pathway and PI3K/AKT signaling pathway. We speculated that the MAPK signaling pathway, the PI3K/AKT signaling pathway, and the targets involved play an important role in the mechanism of TM anti-HCC.

Figure 5 GO and KEGG enrichment analysis.

The GO results are presented as bubble plots. The larger the dot, the more targets are involved. The KEGG results are presented as a chord plot.

Herb-ingredient-target-pathway network

The herb-ingredient-target-pathway network, containing 57 nodes and 384 edges, was constructed (Fig. 6). The purple hexagon node denotes TM, blue V-shaped nodes denote the key ingredients, red circle nodes denote the hub targets, and green diamond nodes denote pathways.

Figure 6 The herb-ingredient-target-pathway network.

The purple hexagon node denotes TM, blue V-shaped nodes denote the key ingredients, red circle nodes denote the hub targets, and green diamond nodes denote pathways.

Molecular docking and MD simulations

Considering the results of KEGG pathway enrichment analysis and network analysis, we selected 12 (TP53, AKT1, MYC, EGFR, MAPK3, VEGFA, CCND1, PTEN, EGF, IL-6, HRAS, HSP90AA1) hub targets to dock with the key ingredients using AutoDock Vina, the PDB IDs and structures of proteins are shown in Table 5. A total of 168 docking results were obtained, of which 139 showed binding energies below −5.0 kcal/mol-1 (Fig. 7). Quercetin, Sonchuside A, ethyl caffeate, taraxinic acid, and Austricin play an important role for TM anti-HCC since they bind well to each target. CCND1, PTEN, HRAS, and HSP90AA1 are the priority targets since their binding energy to each ingredient is lower than −5.0 kcal/mol-1. The binding energies of HSP90AA1 to Austricin, HSP90AA1 to Quercetin, PTEN to Sonchuside A, HRAS to Quercetin, and MAPK3 to Austricin are the five lowest and visualized by PYMOL (Fig. 8). MD simulations of the above five complexes were carried out, to further confirm precise binding mechanisms and interaction stability. RMSD, RMSF, binding free energy, and energy components were reported for individual complexes. The RMSD of HSP90AA1 with Austricin and HSP90AA1 with Quercetin remained stable through the whole simulationG, indicating the high stability of these two complexes (Figs. 9A, 9B). Violent fluctuation of RMSD indicates violent motion of the complex, and conversely, stable motion. RMSF results showed that HSP90AA1 was in a low-flexible stable state after conjugation with Austricin/Quercetin (Figs. 9C, 9D). As shown in Fig. 10, the conformations of the HSP90AA1-Austricin and HSP90AA1-Quercetin complexes were consistent pre and post simulation, with high protein overlap. Austricin sits within the pocket formed by hydrophobic amino acids such as PHE123, VAL121, TYR124, LEU92, LEU89, VAL135, ILE89, LEU88, and TRP147. Quercetin binds hydrophobically to ALA40, THR169, ASN36, PHE123, LEU92, VAL135, MET83, VAL171, TYR124 on HSP90AA1. The hydrogen bonding interaction between ASP78/SER37 on HSP90AA1 and Quercetin contributes to the stability of the complex. The binding free energies and energy components of these two complexes are shown in Table 6.

Table 5 PDB IDs and structures of the proteins for molecular docking.

Target	PDB ID	Structure	
TP53	6SL6		
AKT1	6HHI		
MYC	6G6K		
EGFR	6S9C		
MAPK3	2ZOQ		
VEGFA	3QTK		
CCND1	2W9F		
PTEN	7JVX		
EGF	1JL9		
IL6	7NXZ		
HRAS	7DPJ		
HSP90AA1	4BQG		

Figure 7 The binding free energy of molecular docking.

The results of molecular docking are presented as a heat map. The darker the color, the lower the binding energy and the more stable the binding.

Figure 8 Five best dockings with visualization by PYMOL.

(A) Austricin binds to one residue (TYR139) in HSP90AA1. (B) Quercetin binds to two residues (GLY135, SER522) in HSP90AA1. (C) Sonchuside A binds to four residues (ASN323, ARG173, ARG172, GLN149) in PTEN. (D) Quercetin binds to two residues (ALA146, LYS147) in HRAS. (E) Austricin binds to one residue (MET125) in MAPK3.

Figure 9 RMSD and RMSF of MD simuiation.

(A) The RMSD of HSP90AA1-Austricin remained stable through the whole simulation. (B) The RMSD of HSP90AA1-Quercetin remained stable through the whole simulation. (C) The RMSF of HSP90AA1 when conjugation with Austricin. (D) The RMSF of HSP90AA1 when conjugation with Quercetin.

Figure 10 Pre- and post-MD simulation conformations and specific binding sites of the HSP90AA1-Austricin and HSP90AA1-Quercetin complexes.

(A) Binding mode of HSP90AA1-Austricin. (B) Binding mode of HSP90AA1-Quercetin.

TM inhibits proliferation and HSP90AA1 gene expression of HCC cells

It is observed microscopically that TM-treated HepG2 cells grew slowly and did not adhere firmly to the wall after 24 h of culture compared to the control group (Fig. 11A). CCK8 assay revealed a concentration-dependent inhibition of the proliferation of HepG2 cells in TM treatment. Except for 5 mg/ml, the other different doses of treatment groups (10 mg/ml, 15 mg/ml, 20 mg/ml) showed significant inhibitory effects on the proliferation of HepG2 cells compared to the 0 mg/ml group (Fig. 11B). In addition, it was found that the half-maximal inhibitory concentration (IC50) value of TM on HepG2 cells was 6.911 mg/ml (Fig. 11C). Based on the IC50 value, we used RT-qPCR assay to analyse the regulatory effects of TM on HSP90AA1 gene to validate the in silico results. As shown in Fig. 11D, TM significantly inhibited the mRNA expression of HSP90AA1 (P < 0.05).

Discussion

TM is a widely used herb. Previous studies have shown its treatment potential targeting various malignant tumors, including HCC, but the detailed underlying effector mechanism remains unclear. This study combined network pharmacology analysis and computer virtual verification to provide a specific anti-HCC mechanism of TM, with the aim of providing a basis for further research and clinical application of TM anti-HCC.

In this study, we recognized 22 hub targets and 14 key ingredients by network analysis. Majority of the hub targets belong to MAPK and PI3K/Akt signaling pathways, such as P53, AKT1, MYC, EGFR, MAPK3, VEGFA, EGF, IL-6, RAS. The function of the MAPK signaling pathway is to transmit extracellular signals to the nucleus, thus regulating cell proliferation, differentiation, apoptosis, and autophagy. The MAPK pathway remains activated in approximately 90% of liver cancers, which is known to be the crucial pathway in proliferation and metastasis of carcinoma cells (Moon & Ro, 2021). The MAPK pathway is one of the important mechanisms of sorafenib in the treatment of HCC (Kim et al., 2018). PI3K/AKT, one of the major intracellular signaling pathways, is frequently activated in HCC. It regulates divers cellular functions, such as cell apoptosis, cell migration, angiogenesis. Activation of the PI3K/AKT pathway is significantly associated with reduced overall survival in HCC. There is also crosstalk between the MAPK and PI3K/AKT pathways. MAPK inhibition results in a negative feedback loop that activates the PI3K/AKT pathway (Mirzoeva et al., 2009). Therefore, drugs targeting both the MAPK and PI3K/AKT pathways should be an excellent therapeutic strategy for HCC. Considering the importance of these two pathways, the hub targets associated with them were selected to dock with the key ingredients.

The 14 key ingredients, including Quercetin, Palmitic acid, Linolenic acid, Esculetin, Sonchuside A, Ethyl 4-hydroxyphenylacetate, 1-(3,7,7-Trimethyl-4-bicyclo[4.1.0], hept-3-enyl)ethanone, D-Ascorbic acid, Ethyl caffeate, Caffeic acid, Taraxinic acid, Methyl caffeate, Austricin, and Vanillin, exhibit significant anti-HCC effects. For example, Quercetin induces apoptosis in hepatocellular carcinoma cells in vivo and in vitro by regulating multiple pathways such as PI3K/Akt, MAPK/ERK, and JAK/STAT (Fernandez-Palanca et al., 2019; Ji et al., 2019; Salama et al., 2019; Wu et al., 2019a; Yamada, Matsushima-Nishiwaki & Kozawa, 2021). The increase of metastasis potential of hepatocarcinoma cells usually accompanies the down-regulation of a variety of phospholipid molecules containing palmitic acid. Palmitic acid decreases membrane fluidity of hepatocarcinoma cells by disrupting glucose uptake and lactate production, which impedes the progression of HCC (Lin et al., 2017). Linolenic with anticancer, antioxidant, anti-atherosclerotic, antibacterial, and anti-inflammatory activities realizes the chemoprotective effects via ameliorating the hypoxia microenvironment and regulating mitochondria-mediated apoptotic and anti-inflammatory pathways in diethylnitrosamine induced HCC (Cui et al., 2018; Dubey, Sharma & Kumar, 2019). The cytotoxic effect of Ethyl caffeate on HepG2 hepatoma cells has been demonstrated, but the mechanism remains an open question (He et al., 2009).

Table 6 The binding free energy and energy components of MD simulation.

System name	HSP90AA1-Austricin	HSP90AA1-Quercetin	
ΔEvdw	−33.54 ± 0.97	−30.62 ± 0.82	
ΔEelec	1.33 ± 1.08	−45.59 ± 1.20	
ΔG GB	11.21 ± 1.15	43.64 ± 0.76	
ΔG SA	−3.98 ± 0.05	−5.03 ± 0.06	
ΔG bind	−24.97 ± 0.94	−37.60 ± 0.51	
Notes.

ΔEvdW: van der Waals energy.

ΔEelec: electrostatic energy.

ΔGGB: electrostatic contribution to solvation.

ΔGSA: non-polar contribution to solvation.

Figure 11 Results of cellular experiments.

(A) Microscopic appearance of cells in the control and TM treatment groups after 24 h of cultivation. (B) The results of CCK8 assay and the difference in absorbance value between groups. (C) The dose-inhibition curve of TM at 24 h. (D) Differential expression of the HSP90AA1 gene. **P < 0.01, ****P < 0.0001.

The molecular docking results indicate that Quercetin, Sonchuside A, Ethyl caffeate, Taraxinic acid, and Austricin exhibited superior targeting ability to the hub targets. CCND1, PTEN, RAS, and HSP90 may be critical targets of TM anti-HCC based on binding energy. CCND1, PTEN, and HSP90 are proteins in the PI3K/AKT pathway, PTEN and HSP90 in the upstream, while CCND1 belongs to the downstream effector protein. PTEN is an oncogene with protein phosphatase activity and lipid phosphatase activity, and has been correlated with tumor growth, metastasis and chemoresistance (Alvarez-Garcia et al., 2019). PTEN negatively regulates the PI3K/AKT pathway through dephosphorylation of phosphatidylinositol (3,4,5)-trisphosphate (PIP3) and eventually participates in the modulation of cell proliferation, migration, invasion, and drug resistance during HCC progression (Jiang et al., 2018; Ohta et al., 2015). HSP90, a member of heat shock proteins, plays an important role in the assembly, manipulation, folding and degradation of its client proteins. HSP90 is closely associated with tumors due to its regulation of many client proteins that are proto-oncogene products or important signal transduction factors in tumorigenesis (Shi et al., 2020). Previous studies have confirmed that HSP90 is highly expressed in HCC and promises to be a new diagnostic marker and therapeutic target (Xu et al., 2017). Activation of AKT by HSP90 regulates cellular autophagy and apoptosis mediated by the PI3K/AKT pathway, which is associated with tumor recurrence and drug resistance (Hu et al., 2015). CCND1 is an important cell cycle regulatory protein that promotes the proliferation of cancer cells by forming a cell cycle-dependent complex with cyclin-dependent kinase 4 (CDK4), which contributes to the development of many tumor diseases, including HCC (Tashiro et al., 2003). Activation of the PI3K/AKT pathway induces the accumulation of CCND1, which consequently promotes abnormal hepatocyte proliferation and carcinogenesis (Wu, Lan & Liu, 2019b). RAS is a protein on the MAPK signaling pathway, through which crosstalk between MAPK and PI3K/AKT pathways is achieved. PI3K can be activated by RAS superfamily of GTPases, the interaction of RAS and PI3K plays a key role in promoting tumor formation and maintenance in RAS-driven tumors (Siempelkamp et al., 2017). RAS significantly promotes cell proliferation in HCC by activating both MAPK and PI3K/AKT pathways (Shen et al., 2016). Therefore, the targets derived from the MAPK and PI3K/AKT pathways of the key ingredients are important for the prevention and treatment of HCC.

The HSP90 protein consists of an N-terminal domain, an middle domain and a C-terminal domain. The N-terminal domain is a dimeric structure containing an ATP binding site. The middle domain and the C-terminal domain are the binding regions for substrate proteins and helper molecular chaperones. HSP90 are characterized by a distinct ‘Bergerat fold’ in the N-terminal ATP-binding domain (Chene, 2002). Occupancy of this pocket by small molecule inhibitors inactivates HSP90 chaperone function. It is well established that N-terminal inhibition of HSP90 is effective in inhibiting tumour cell activity in vitro and 18 N-terminal inhibitors of HSP90 have been developed with clinical evaluation (Patel et al., 2013). In this study, we obtained a stable binding model of Austricin/Quercetin to the N-terminal ATP-binding domain of HSP90 through molecular docking and MD simulations. Therefore, it is speculated that the N-terminal inhibition of HSP90 by Austricin/Quercetin may contribute to the anti-HCC of TM.

Conclusion

In summary, the active ingredients of TM and their molecular targets in HCC were successfully unveiled by network pharmacology, molecular docking, MD simulation, and cellular experiments. A total of 14 key ingredients and 22 hub targets were identified. MAPK and PI3K/AKT signaling pathways were found to be potentially primarily responsible for TM anti-HCC. The interaction between Austricin/Quercetin and HSP90 is important for the mechanism of TM anti-HCC. TM may serve as a promising complementary and alternative drug for HCC but needs further in vivo/in vitro experiments. This study provides a holistic view of the potential pharmacological mechanisms for TM anti-HCC, establishing the foundations for further study on the optimization of experimental designs for more reliable results.

Supplemental Information

Supplemental Information 1 The targets of HCC

Click here for additional data file.

Supplemental Information 2 The targets of Taraxacum Mongolicum

Click here for additional data file.

Supplemental Information 3 GOBP

Click here for additional data file.

Supplemental Information 4 GOCC

Click here for additional data file.

Supplemental Information 5 GOMF

Click here for additional data file.

Supplemental Information 6 KEGG

Click here for additional data file.

Supplemental Information 7 The data of molecular dynamics simulations

Click here for additional data file.

Supplemental Information 8 The components of Taraxacum Mongolicum from HERB database

Click here for additional data file.

Supplemental Information 9 Comparison of the intersection targets with differentially expressed genes in the TCGA Liver Cancer Dataset

Click here for additional data file.

Additional Information and Declarations

Competing Interests

Author Contributions

Data Availability

The authors declare there are no competing interests.

Yanfeng Zheng conceived and designed the experiments, performed the experiments, analyzed the data, prepared figures and/or tables, authored or reviewed drafts of the article, and approved the final draft.

Shaoxiu Ji performed the experiments, analyzed the data, prepared figures and/or tables, and approved the final draft.

Xia Li performed the experiments, analyzed the data, prepared figures and/or tables, and approved the final draft.

Quansheng Feng conceived and designed the experiments, authored or reviewed drafts of the article, and approved the final draft.

The following information was supplied regarding data availability:

The raw measurements are available in the Supplemental Files.

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
