# Peer review of "Active ingredients and molecular targets of Taraxacum mongolicum against hepatocellular carcinoma: network pharmacology, molecular docking, and molecular dynamics simulation analysis"

_PeerJ, doi:10.7717/peerj.13737_

## Round 0.1 · original submission · Major Revisions

Reviewers have raised some serious concerns which requires substantial and thorough revision to appreciate the quality for publication in PeerJ. MAJOR revision is suggested. Therefore, authors are requested to revise their manuscript in light of reviewers comments and resubmit accordingly.
Furthermore, thorough English editing is required. Please revise the manuscript taking help from a colleague who is proficient in English and familiar with the subject matter, who can review your manuscript, or contact a professional editing service to review your manuscript.

·

Basic reporting

Few figures need to be improved. Such as Figure 2 and 4.
1. If possible, the figure 4 could be edited for better view with clear content of labels.

Experimental design

2. In line no. 133, the authors have mentioned “A binding free energy less than zero indicates that the ligand binds to the receptor spontaneously”. Do the ligands bind repeatedly with the receptor or any spontaneous process of the thermodynamics occurs? Revise the statement accordingly. Please refer the below article for detailed information
a. Insights into Protein-Ligand Interactions: Mechanisms, Models, and Methods. International journal of molecular sciences, 17(2), 144. (2016) https://doi.org/10.3390/ijms17020144

Validity of the findings

4. The authors have selected only five best complex for molecular dynamics studies based on their binding energies. If possible display the remaining other targets with ligands not included in further analysis with binding energies in supplementary. It could be helpful for the follow-up studies

Additional comments

3. In line 174, “After the duplicates were removed, one thousand two hundred and twenty-three targets were obtained”, the authors could mention the same in numbers/integers and maintain uniformity.

·

Basic reporting

.

Experimental design

.

Validity of the findings

.

Additional comments

The manuscript is novel, well written and the results are interesting, however, the major point that I have a concern regard are;
1- Experimental in vitro enzyme assay for the best selected target such as Hsp90 , PI3K , ....
one enzyme assay for one or two representative component of the plant will validate the in silico results.
2- Cell line inhibition measurement of the most representative component of the plant.
3- correlation of enzyme assay inhibition with cell line inhibition for both pure compounds and herbal extract will validate the finding more.
4- it is not clear if the anti-tumor effect of the herb extract is due to a principal compound or due to cumulative effect of the herb components, so I think it is required from the authors to clarify this point.
5- Statistical validation of the method used, use a reference or standard to compare with is essential.
6- I think addition of the scheme or flowchart for the sequence of the work will help understanding the logic of the manuscript, specially the idea seems novel.
7-Validation of the docking process by redocking of cocrystallized ligand and find RMSD values.

Reviewer 3 ·

Basic reporting

This paper addresses an interesting topic: to shed some light on the mechanisms of action of the herb Taraxacum mongolicum. The authors use the principal databases and software for data analysis and predictions to their scope and they find two possible active ingredients (Austricin and Quercetin) that probably interact with the protein encoded by the gene HSP90AA1 (Heat shock protein HSP 90-alpha).
The paper is well organized, includes sufficient introduction and background and the methods are described precisely.

Experimental design

The research question is well-defined and meaningful.
The investigation is all in silico, but it is well performed.
The methods are described precisely.

Validity of the findings

All the data have been provided.
The results are quite well presented, however they could be better described and discussed. In particular, it should be shown or described where the molecules interact with HSP60 with respect i.e. to ATP binding site in order also to speculate on possible consequences of the binding. Moreover, at least in the discussion section, the gene symbols should not be used to identify the proteins. Since there are no lab experiments confirming the in silico results, It is also necessary to maintain doubt sentences.
The figures need captions and some of them do not appear of sufficient resolution.

Additional comments

The English language should be improved. Some examples where the language could be improved are included in the PDF revised file.
I have some minor suggestions to make, therefore, please, check the PDF file.

Annotated reviews are not available for download in order to protect the identity of reviewers who chose to remain anonymous.

---

## Round 0.2 · accepted · Accept

This manuscript now can be considered for publication, as the authors have adequately addressed all the concerns raised by the reviewers.

Reviewer 3 ·

Basic reporting

The authors have carefully revised the original manuscript. Moreover, they have performed lab experiments in order to support their findings. I have no other comments.

Experimental design

no comment

Validity of the findings

no comment